# Deep splicing plasticity of the human adenovirus type 5 transcriptome drives virus evolution

I'ah Donovan-Banfield[1], Andrew S. Turnell [2], Julian A. Hiscox[3], Keith N. Leppard[4] & David A. Matthews [1⊠]

Viral genomes have high gene densities and complex transcription strategies rendering transcriptome analysis through short-read RNA-seq approaches problematic. Adenovirus transcription and splicing is especially complex. We used long-read direct RNA sequencing to study adenovirus transcription and splicing during infection. This revealed a previously unappreciated complexity of alternative splicing and potential for secondary initiating codon usage. Moreover, we find that most viral transcripts tend to shorten polyadenylation lengths as infection progresses. Development of an open reading frame centric bioinformatics analysis pipeline provided a deeper quantitative and qualitative understanding of adenovirus's genetic potential. Across the viral genome adenovirus makes multiple distinctly spliced transcripts that code for the same protein. Over 11,000 different splicing patterns were recorded across the viral genome, most occurring at low levels. This low-level use of alternative splicing patterns potentially enables the virus to maximise its coding potential over evolutionary timescales.

[1] Department of Cellular and Molecular Medicine, School of Medical Sciences University Walk, University of Bristol, Bristol BS8 1TD, UK. [2] Institute of Cancer and Genomic Sciences College of Medical and Dental Sciences University of Birmingham Edgbaston, Birmingham B15 2TT, UK. [3] Department of Infection Biology, Institute of Infection and Global Health, University of Liverpool, ic2 Building, Liverpool L3 5RF, UK. [4] Life Sciences University of Warwick Coventry, Coventry CV4 7AL, UK. ⊠email: d.a.matthews@bristol.ac.uk

Forty years ago eukaryotic mRNA splicing was discovered by hybridising adenovirus mRNA to adenoviral DNA (refs. [1,2]). Over time, study of the adenovirus transcriptome revealed an increasingly complex repertoire of alternative promoter, splicing and polyadenylation usage, making the adenovirus transcriptome one of the most complex of any virus studied to date.

The viral linear double-stranded DNA genome of ~36,000 bp, is delivered to the host cell nucleus rapidly after virus entry, where mRNA is transcribed from both DNA strands using host cell RNA pol II. Transcription from ten different transcription units proceeds in early, intermediate or late temporal classes, as defined relative to the onset of viral DNA replication[3]. There are early (E1a, E1b, E2, E3 and E4) and intermediate (IVa2, pIX and L4) transcription units, as well as the major late transcription unit (MLTU) driven by the major late promoter (MLP) that produces transcripts for most of the virus's structural proteins. Except for pIX mRNA, all adenovirus transcripts are spliced. The MLTU itself has over 20 distinct alternative splice sites and five distinct polyadenylation sites giving rise to five different classes of mRNA L1–L5. There are three non-coding exons at the 5′ of all known late mRNA (known collectively as the tripartite leader, TPL) as well as three non-coding exons present in varying amounts on a subset of MLTU mRNA (known as the x-, y- and z-leaders)[4,5]. Finally, there is the i-leader exon, which is infrequently included between the second and third TPL exons, and codes for the i-leader protein[6]. Thus, the MLTU produces a complex array of mRNA with diverse 5′-UTRs[5], some of which are known to influence mRNA stability, spliced onto different 3′ coding exons grouped into five different 3′-end classes.

Short-read deep sequencing has created a high-resolution image of the adenovirus transcriptional landscape[7,8]. However, there are difficulties determining how multiple splicing events are linked on a particular mRNA. For example, when i-leader exon is included in an MLTU mRNA, it is unclear which of the possible MLTU 3′ exons it has been added to. Secondly, in short-read sequencing the mRNA is fragmented, reverse transcribed into cDNA, then amplified by PCR before sequencing using a second PCR-based method. These steps could introduce bias confounding qualitative and quantitative analysis. Nanopore-based direct RNA sequencing (dRNA-seq) avoids these issues because individual mRNA molecules are introduced, poly-A tail first, into the nanopore with their sequences read as they pass through the pore. This provides a direct and complete record of the exons present in any given mRNA without reverse transcription or amplification steps. dRNA-seq has recently been used to examine the transcriptome of human cells[9,10], coronavirus[11] and herpes simplex virus[12], revealing greater complexity than previously appreciated and underlining the power of this approach to improve our understanding of transcriptomes, even in well-studied systems.

Here, dRNA-seq was used to examine the transcriptome of human adenovirus type 5 during an infection, revealing an unbiased picture of the viral transcriptome in unparalleled detail; almost 1.2 million viral transcripts over three time points. The complexity of viral mRNA species was much larger than expected from previous analyses. Key to this deeper understanding was the development of an ORF-centric approach to classification of the data. We provide the raw data, software tools and processed data enabling other researchers to interrogate this rich dataset. This analysis cements adenovirus's reputation as probably the most transcriptionally complex virus examined to date, with respect to splice and polyadenylation site usage.

## Results

### Overview of sequencing data and analysis pipeline outputs.
The proportion of sequences identified as adenovirus transcripts increased from around 2% at 16 h post infection (h.p.i.) to 12% at 24 h.p.i., and just over 47% by 48 h.p.i. (Supplementary Table 1). This increase is comparable to that previously reported for adenovirus infection of another primary human lung line IMR-90 (ref. [7]; i.e., 2.7% reads mapped to adenovirus at 12 h.p.i rising to 58.2% by 36 h.p.i). Reassuringly, the longest reads detected, average read lengths, dominant read length sizes and longest transcript mapping to the virus did not markedly change over time (Supplementary Table 1).

A key output of our pipeline is an overview showing the dominant transcript that codes for each identified protein compared to the classical map of adenovirus transcription. Transcript maps for the major expressed ORFs at each time point correlated well with the classical view of the viral transcriptome (Fig. 1a, b). Moreover, a broad overview (Supplementary Table 1) or a protein by protein breakdown (Fig. 2) of transcripts expressing known adenovirus ORFs confirmed the expected pattern of increasing dominance of late protein expression as the infection progresses[13].

Thus, our ORF-centric approach generates an accurate overview of the dominant transcriptional output based solely on the location of known ORFs within the viral genome.

### Nanopore-inferred transcription start sites and transcriptional termination sites.
The promoters and polyadenylation sites on the adenovirus genome have been mapped previously with high accuracy. We noticed (Supplementary Table 2, Supplementary Data 1–3) that for each promoter, the apparent transcription start sites (TSS) indicated by the 5′-ends of the majority of nanopore transcripts in a group was ~8–15 nucleotides downstream of the established location. This kind of discrepancy has been reported previously for both human and herpesvirus data[9,12,14]. Therefore, in order to allow correct prediction of the protein-coding potential of a transcript, each adenovirus-mapped pseudo transcript sequence was programmatically extended by 10 nt at its 5′-end prior to conducting ORF analysis.

In addition to the large numbers of transcripts detected with 5′-ends corresponding to previously mapped adenovirus TSSs, smaller numbers of transcripts were found that had 5′-ends located elsewhere. The numbers of these mapping to a location broadly correlated with the overall level of transcription through that region of the genome. We interpret these as coming from RNA breakage prior to sequencing or incomplete sequencing from the 3′-end.

Mapping the 3′-ends of the transcripts, we observed multiple transcriptional termination sites (TTSs) for each major class of transcript in addition to those previously defined. Collectively, these represent a large fraction of the viral polyA events. For example, transcripts with 3′-ends mapping collectively to the known L1–L5 polyA sites represented only around half of the MLP-derived mRNAs. Our analysis was confined only to transcripts, which had confirmed polyA tails at least 20 nt long on the 3′-end, hence it can be excluded that these variant 3′-ends represented fragmentations prior to sequencing of molecules that were actually correctly polyadenylated at the known sites.

We also examined the mapping alignment outputs from minimap2 for the presence and scale of soft clipping at the 5′- and 3′-ends using an in-house script. Soft clipping is a standard feature of sequence alignment software and refers to the practice of effectively ignoring a short number of nucleotides at the 5′- or 3′- end of a sequence read to allow the rest of the sequence to be aligned to the target genome. We observed that soft clipping of transcript ends was widespread only at the TTS (Supplementary Table 2). Most transcripts were not soft clipped at the 5′ TSS and when they were, the average soft clip size was typically <10 for 5′ TSS assignments with a typical modal of one or two nucleotides. By comparison, TTS mapping assignments were almost always

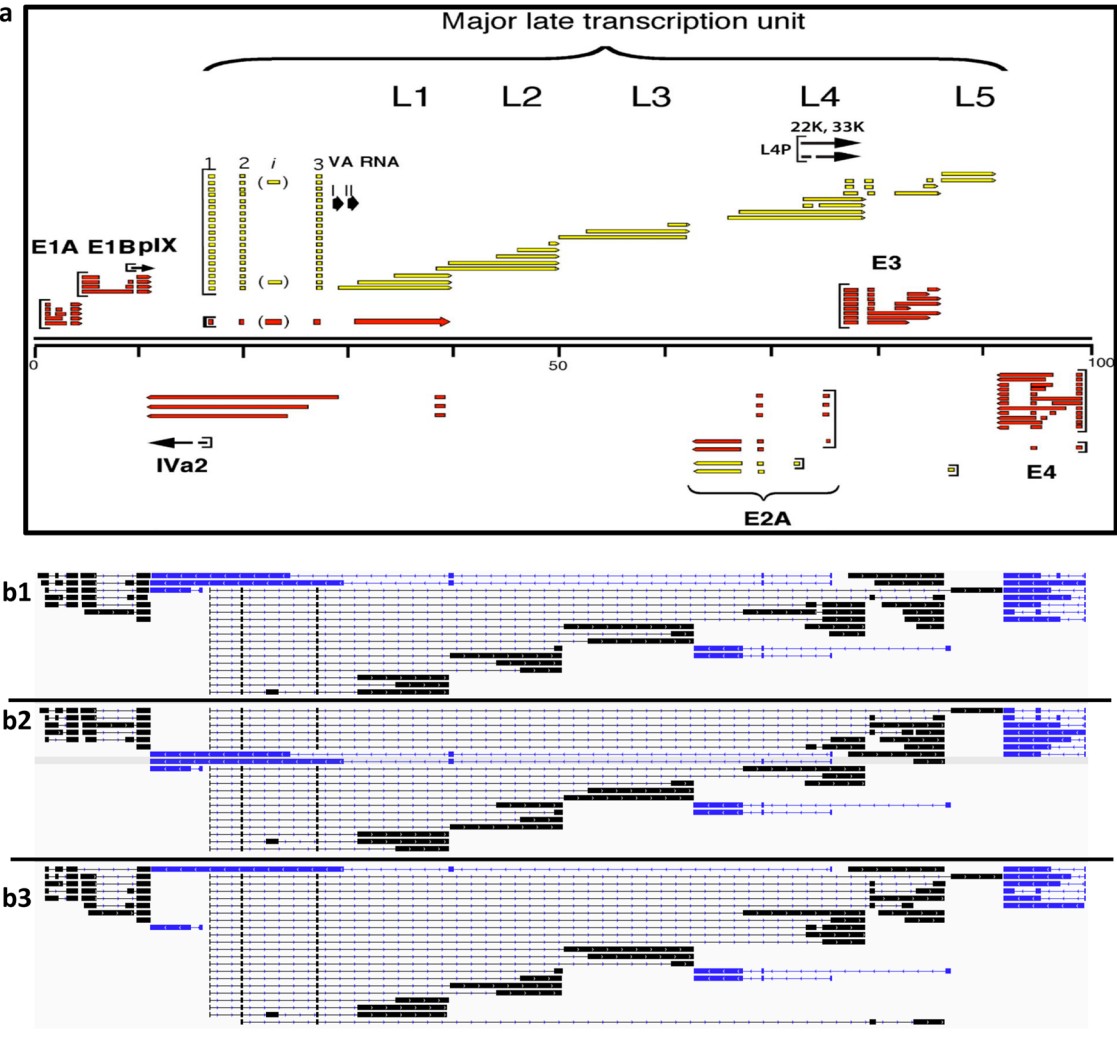

**Fig. 1 Transcription map overviews. a** Classical transcription map of adenovirus type 5. Transcripts shown above the genome are coded for on the top strand from left to right, while those below the genome are coded for on the bottom strand, transcribed from right to left. Genes are colour coded in red for early genes and yellow for late genes, denoting if their expression is predominantly before or after the onset of adenovirus DNA replication. Square brackets indicate classical TSS. Genes pIX and IVa2 are shown in black as they are classed as intermediate in their expression timing. The major late transcripts are further broken down into L1–5 according to their shared polyadenylation sites. **b** Transcript maps representing 16, 24 and 48 h post infection (b1, b2 and b3, respectively) derived from nanopore sequencing data. Each map shows the dominant transcript coding for each of the known adenovirus proteins. The black rectangles represent exons from the top strand of the genome, and blue rectangles represent exons coded by the bottom strand.

clipped to some degree and the average was >15 for 3′ TTS assignments with a typical modal of 2+ nucleotides.

**Early regions 1a and 1b**. Our analysis of E1a and E1b mRNAs revealed substantial numbers of transcripts that conformed to the known mRNAs (refs. [15–19]). Understanding the protein-coding potential of the detected E1b mRNAs is complicated by two factors: the initiating AUG for the 5′ proximal ORF in this region (E1b19k) is very close to the published E1b TSS, and there is a second protein (E1b55k) whose expression arises from a downstream initiating AUG (ref. [20]; Fig. 3). Very few transcripts that could code for E1b19k were detected in our data before programmatic extension of the 5′-end (above), after which the vast majority of detected E1b transcripts coded for this protein. This intervention therefore converted apparently E1b55k mRNAs into E1b19k mRNAs. It is generally understood that translation initiation at the E1b19k AUG is inefficient, leading to the significant expression of E1b55k from these mRNAs as a group but it cannot be excluded, given the presence in our data of mapped 5′-ends lying downstream of the

E1b19k AUG, that some of these RNAs genuinely initiate at these locations and encode E1b55k as the 5′ proximal ORF.

A small number of non-classical transcripts were also present that started at the classical E1a and E1b TSSs, but extended beyond the normal TTS. These were further spliced before terminating at TTSs downstream of the usual E1a or E1b TTS, some extending to the MLTU polyadenylation sites (L2–L5) or to the late E3 TTS (Fig. 3b). Even though in each case we detected less than ten sequence reads, this illustrates the potential for E1a/E1b transcripts to extend beyond their normal TTSs, connecting together a wide range of exons.

**Early region 2**. This region codes for three classical proteins, pTP, Ad-pol (collectively, E2B proteins) and DBP (the E2A protein) from two promoters: E2-early and E2-late[17,18,21]. The RNAs encoding E2B proteins also share their TTS with the IVa2 gene[22]. A further gene known as UXP (ref. [23]) has a shared TTS with the transcripts for DBP. As expected from the previous data, the levels of transcript for pTP and Ad-pol were very low (fewer than five transcripts),

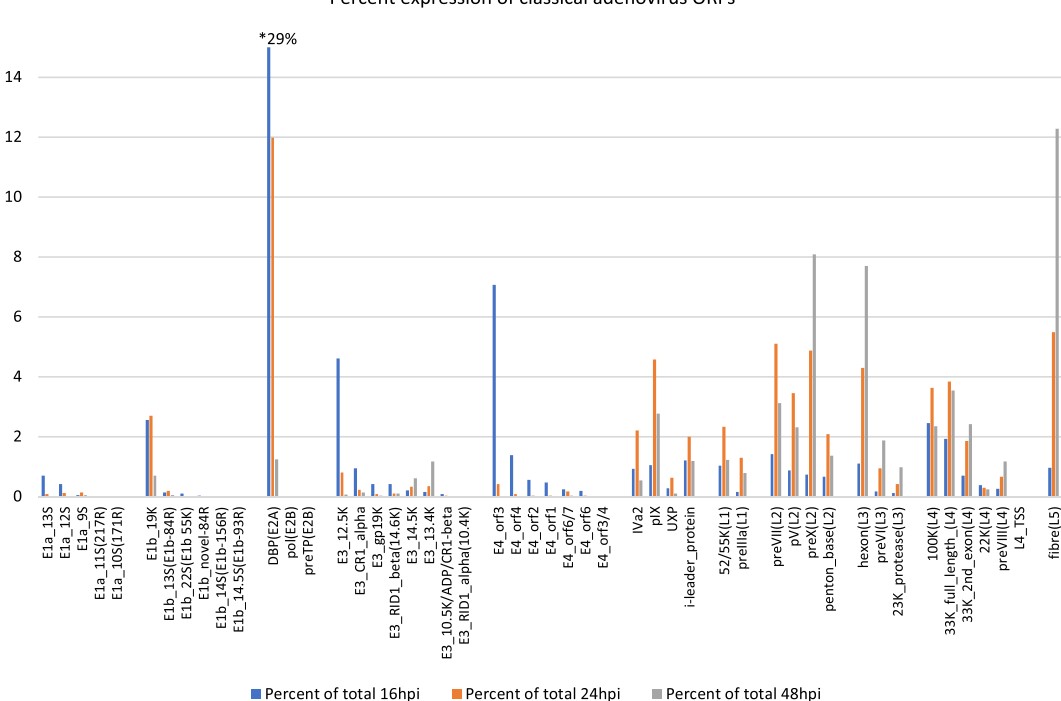

**Fig. 2 Overview of change in expression of known adenovirus ORFs over time.** The percent of transcripts that contain the indicated ORF as the most 5′ ORF at each time point is indicated. Note that these will not add up to 100% as not all transcripts meet this criterion as discussed in the main text. In addition, over the three time points 11% (16 h.p.i.), 13% (24 h.p.i.) and 16% (48 h.p.i.) of transcripts do not code for any known adenovirus protein. To aid clarity, the percent values on the chart are capped at 15%; the value for DBP expression at 16 h post infection is 29%.

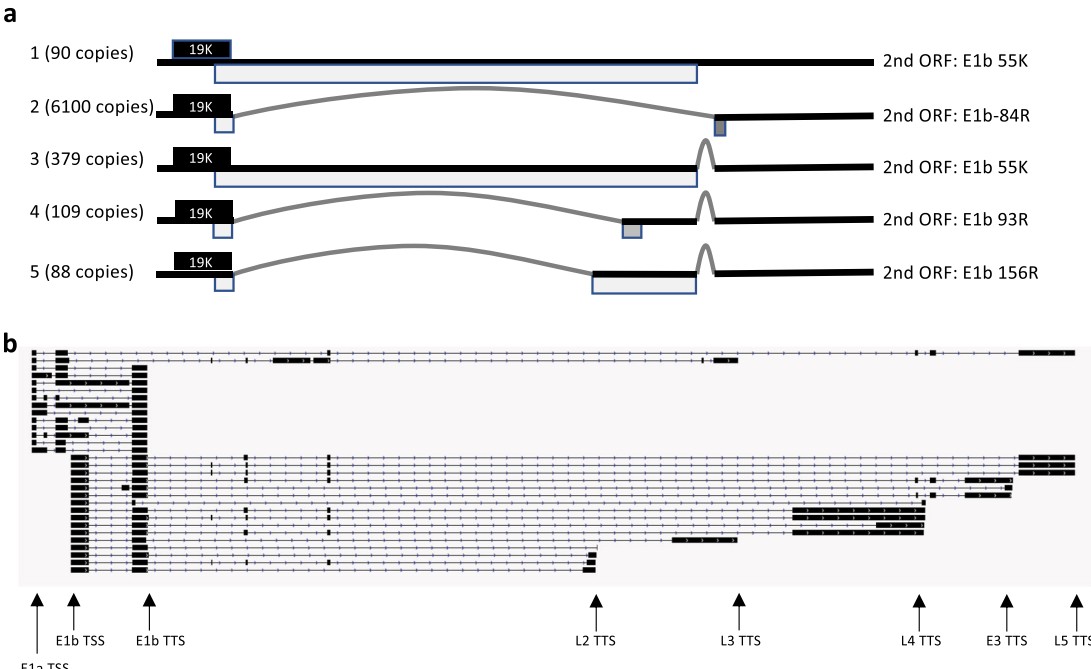

**Fig. 3 E1 region transcripts. a** The coding potential of classical E1b transcripts (TSS at 1705 and TTS at 4073). A solid line illustrates the transcript structure (curved sections indicate introns) with boxes showing the encoded ORFs when appropriately spliced. The black ORF is the E1b19K protein, which is the 5′ most ORF on all classical E1b transcripts. To the left of each transcript is noted how many copies were seen cumulatively across all three time points. To the right is indicated which known E1b protein is coded for by the second initiating AUG. **b** The structure of rare transcripts that initiate at the E1a or E1b TSS but continue beyond the usual E1a or E1b TTS (image generated by IGV viewer). Black boxes indicate exons joined by fine lines with arrows. Each transcript structure shown is unique and in each case is evidenced by one nanopore transcript. The locations of relevant TSS and TTS are shown for orientation, except the normal E1a TTS that is 100 nt upstream of the E1b TSS and is omitted for clarity.

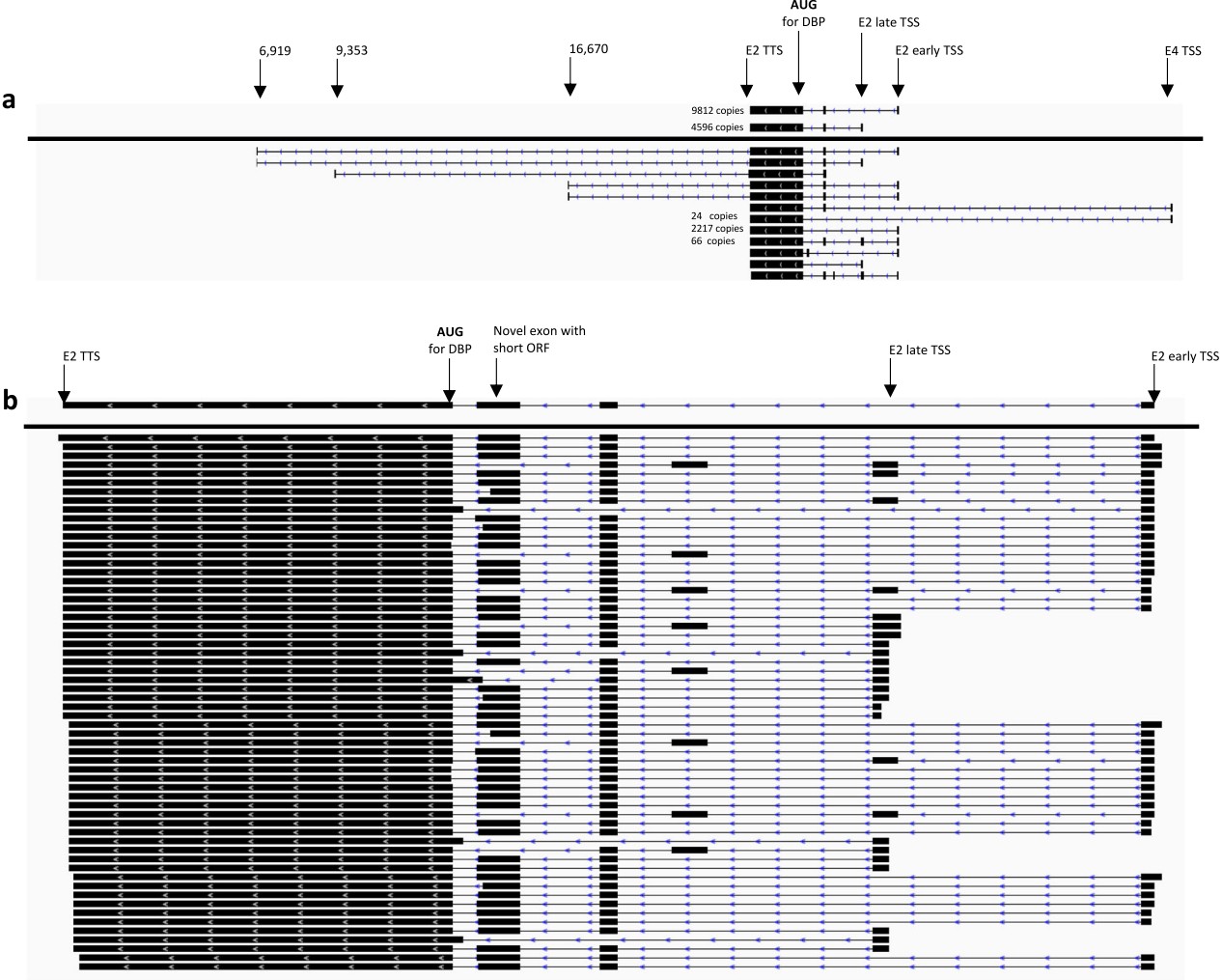

**Fig. 4 E2 region coding for DBP. a** Each line represents a transcript type coding for DBP, with exons represented as black boxes and the number of transcripts of each type detected across all time points indicated, where more than one transcript was sequenced in that transcript group. All the transcripts have right to left polarity. The transcripts above the dividing line represent the classical DBP-coding transcripts starting at the E2-early or E2-late TSS, while below the line is a representative sample of non-classical DBP transcripts also detected. These exemplify exons missing, novel exons included and transcription extending beyond the normal TTS. **b** Novel E2A region transcripts encoding DBP only if the 5′ proximal AUG is skipped. The transcript above the dividing line is the dominant example of this; almost 300 individual transcripts belong to this transcript group. The examples shown below range in abundance from 1 to 240 transcripts observed in each transcript group.

whereas mRNA levels for DBP were very high and were detected coming from both E2-early and E2-late promoters with the E2-early promoter always dominant. We also detected DBP-coding transcripts at all time points driven by the E4 promoter and a small number of individual transcripts that extended beyond the normal TTS (Fig. 4a). Importantly, we detected previously unreported DBP-coding transcripts, including one group without the usual second exon (nt 24,745–24,668) and another with a previously undescribed exon (nt 25,947–25,839; Fig. 4a). We also identified a substantial number of spliced transcripts (with copy numbers < 100) that can only code for DBP if the 5′ proximal AUG is skipped (Fig. 4b). In the dominant example of this, an additional previously unreported exon (24,331–24,150/47) containing a short ORF is incorporated upstream of the DBP-coding exon.

**Early regions 3 and 4.** The E3 region is nested within the MLTU, encoding proteins that subvert the immune response[24]. The E3 promoter dominates E3 transcription at early times post infection, while E3 transcripts driven by the MLP dominate

at later times[25]. In addition, there are two different E3 TTSs (ref. [26]).

Transcripts were detected coding for known E3 proteins coming from the E3 TSS nt 27,572 or the MLP at nt 6051 (Fig. 5a). Except for the 12.5K protein, E3 protein expression relies on translation initiating at the second AUG in an mRNA unless the transcript starts at the MLP and splicing puts the E3 ORF 5′ most on the finished transcript. Our data support the concept that E3 expression is highly complex and provides evidence for rare transcripts clarifying aspects of E3 gene expression. For example, expression of E3 gp19K protein could only previously be explained by translation (re-)initiation at the downstream or second ORF on the E3 Cr1-alpha transcript, something that is known in other viral systems[27]. While this may occur, we detect transcripts from MLP whereby the TPL is directly spliced onto the gp19K ORF (Fig. 5b). Indeed, splicing from the TPL directly to splice acceptor sites just upstream of each of the relevant E3 ORFs is common[28], was reflected in our data, and may be one strategy used by the virus to ensure all E3

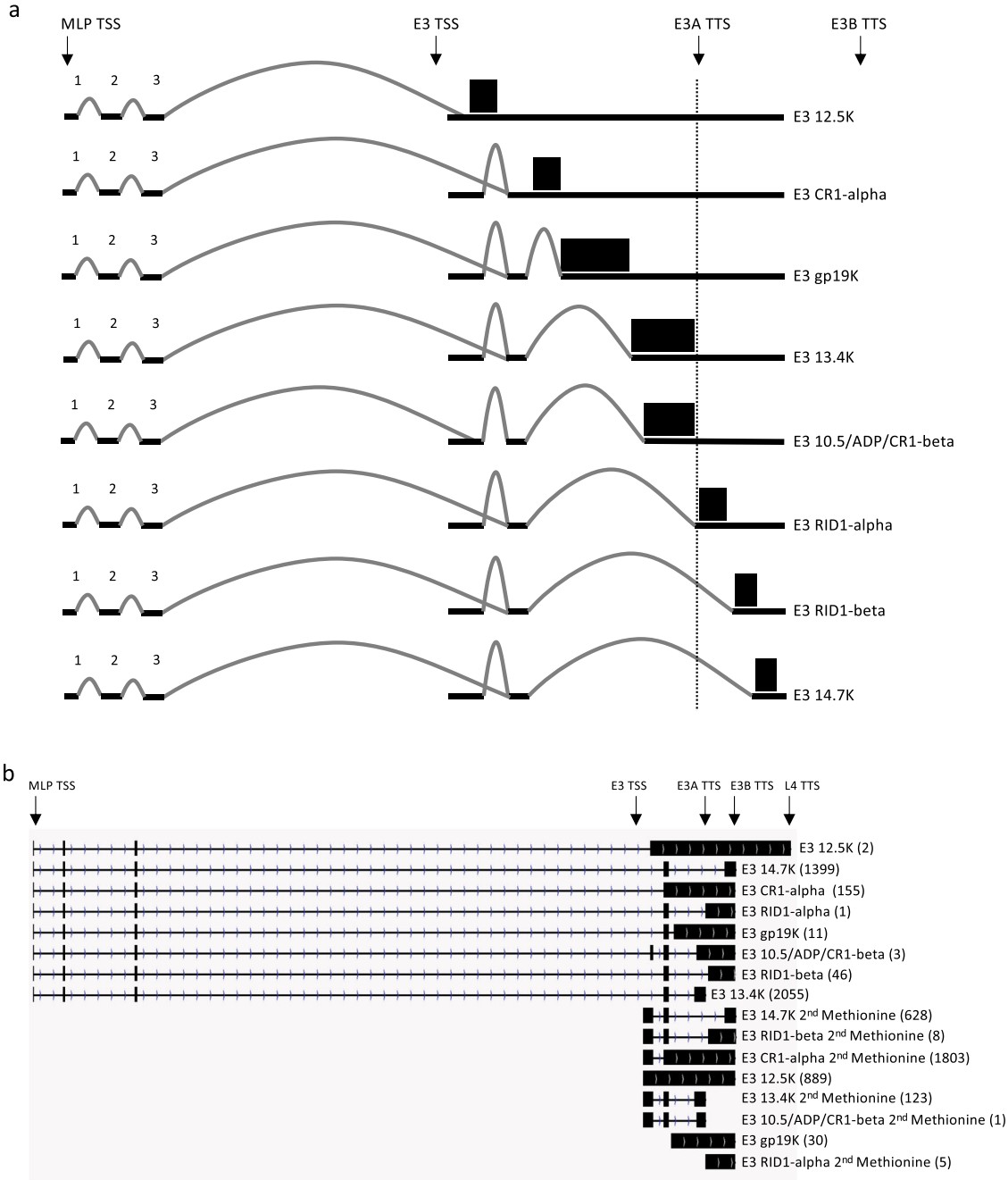

**Fig. 5 E3 transcripts. a** Coding and splicing potential of the E3 region: solid lines represent transcripts, potential splice events are indicated with a curved grey line with black boxes, representing ORFs with the ORF name indicated on the right. This schematic shows the two main promoters that drive E3 expression (MLP and the E3 promoter) and the two main transcription termination sites (E3A and E3B). Note that the initiating AUG for the E3 12.5K protein is present in every E3 transcript originating from the E3 TSS, thus on these transcripts other ORFs can only be expressed if the 5′ most AUG is skipped. **b** shows two representative transcripts for each classical E3 ORF. One representative is the most abundant transcript group initiating at the MLTU TSS, and the second is the most abundant transcript group initiating at (or as near as detected) to the known E3 TSS. In each case, after the indicated ORF name, the number of observed sequence reads that fit this transcript group across all three time points is shown in brackets. Where an ORF has "2nd Methionine" added to the name it indicates that, for this transcript, the 5′ proximal ORF does not code for a known adenovirus protein.

proteins are expressed. Considering E3 transcripts in our data that arose from either the E3 promoter or the MLP as a whole, we identified all known E3 transcript classes. Similar to the other transcription units, there were a large number of non-classical transcripts involving splice events and exon combinations that have not been reported before (Supplementary Data 4).

The E4 region data were dominated by high-copy number transcripts coding for known E4 proteins, but with a large number

of low-copy number alternatives (Supplementary Data 4). Also, there were one or two examples of transcripts that extend beyond the normal E4 TTS and terminate instead at the E2A TTS (Supplementary Data 4). It had been previously reported that there are between 18 and 24 different mRNA produced by the E4 region[29], whereas we observed 39 different splice patterns of transcripts starting at/near the E4 promoter and terminating at or before the E4 TTS (Fig. 6). Again, we observe multiple transcripts

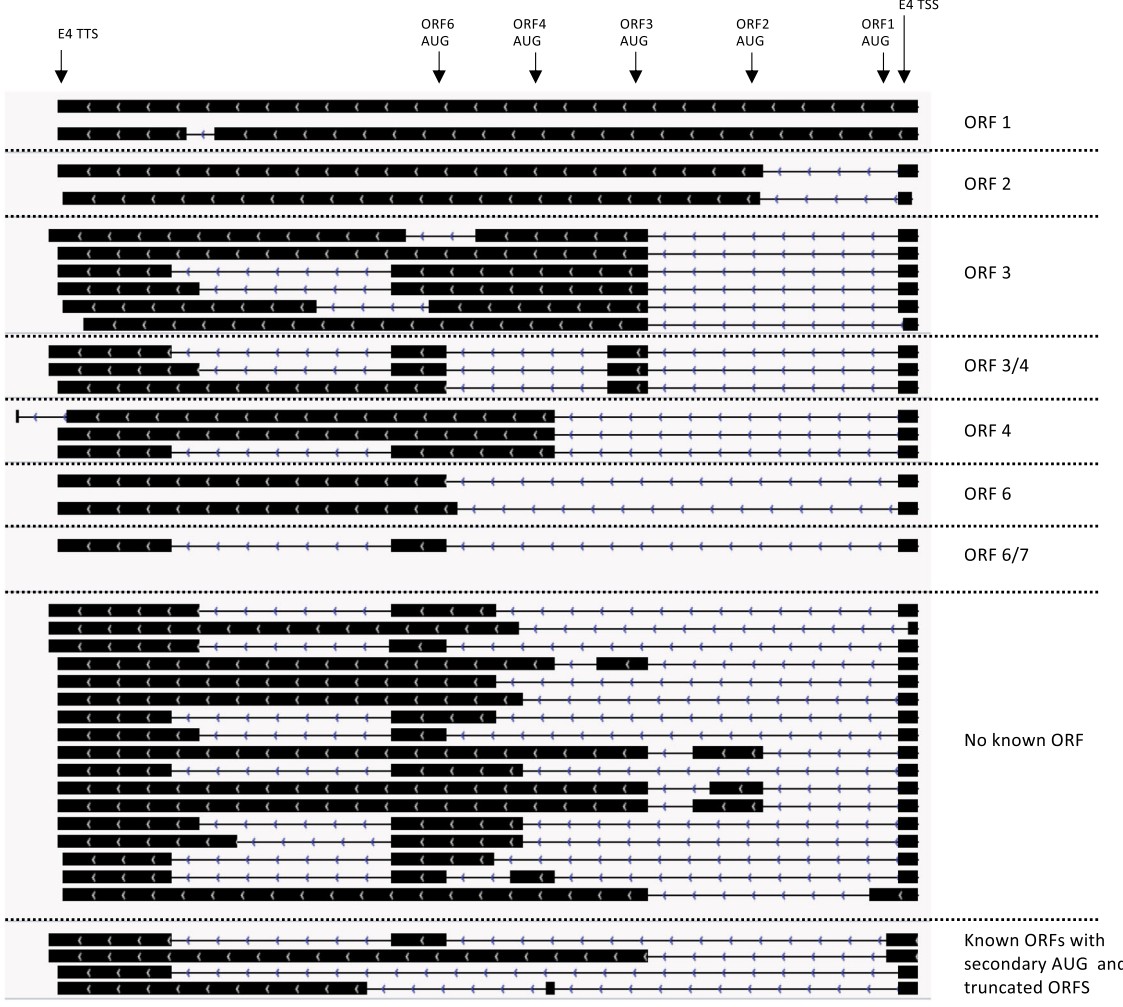

**Fig. 6 E4 transcripts.** An IGV viewer image of transcripts that have an E4 TSS and a TTS within 50 nt of the classical E4 TSS/TTS. These are further characterised according to whether they code for a known E4 ORF as indicated in the diagram. In addition, we show transcripts that code for ORFs that are unknown (labelled as no known ORF). Finally, we show the structure of a small number of rare transcripts (<3 per transcript group) that have a known ORF but only as the second ORF on the transcript or the known ORF is truncated. Also indicated are the locations of the start codons for the known E4 ORFs.

coding for the same ORF as well as numerous transcripts that apparently do not code for any known proteins. This makes comparison with previous RNAse protection assay data, for example, highly problematic[30].

**Late regions 1–5.** mRNAs from MLTU regions L1–L5 code for most of the viral structural proteins and at late time points these transcripts form the bulk of expressed mRNA (Fig. 7a). Overall, the coding potential of the late mRNAs detected was as expected, with L4 transcripts appearing earlier than L2, L3 and L5 (ref. [31]). However, the levels of transcript for each protein did not reflect the stoichiometry of those proteins within the virion. For example, there are 20-fold more copies of hexon polypeptide than fibre polypeptide in each virus particle[32], whereas more fibre (12%) than hexon (7.7%) coding transcripts were detected at 48 h.p.i. (Table 8). There was also a surprising dominance of transcripts coding for preX (also known as preMu), with 8% of the transcripts at 48 h coding for this protein. PreX is proteolytically matured by the viral 23K proteinase post assembly[33,34]; the mature pX associates with viral DNA in the particle alongside viral proteins pV and pVII, providing histone-like functions. There are ~150 copies of pV, ~300

copies of pX and ~500 copies of pVII in a virus particle[35], which is at odds with the observed transcript abundance ratio (2.3% pV, 3.1% preVIII and 8% for preX), suggesting either that the abundance estimate for pX in particles is inaccurate, or that more is made than is needed or that preX/pX proteins have additional roles beyond viral DNA packaging.

A large number of transcripts were spliced from the TPL onto the second exon of the 33K gene within the L4 polyadenylation unit (Supplementary Table 3). These would in principle code for a 13-residue peptide (which we have termed 33K 2nd exon) from the 5′ proximal ORF, which has not been detected to date (Fig. 7b). However, the next ORF in these RNAs encodes preVIII—the precursor to a structural component of the capsid. Similar to other major late proteins, we would expect preVIII expression from transcripts that splice the TPL onto a splice site just upstream of the preVIII ORF. However, there were four times as many '33K 2nd exon' transcripts compared to pVIII-proximal coding transcripts, suggesting that (as in the E1b and E3 regions) there may be an expanded role for non-canonical ribosomal scanning/initiation in controlling adenovirus major late protein expression beyond the ribosome shunting process already reported[36].

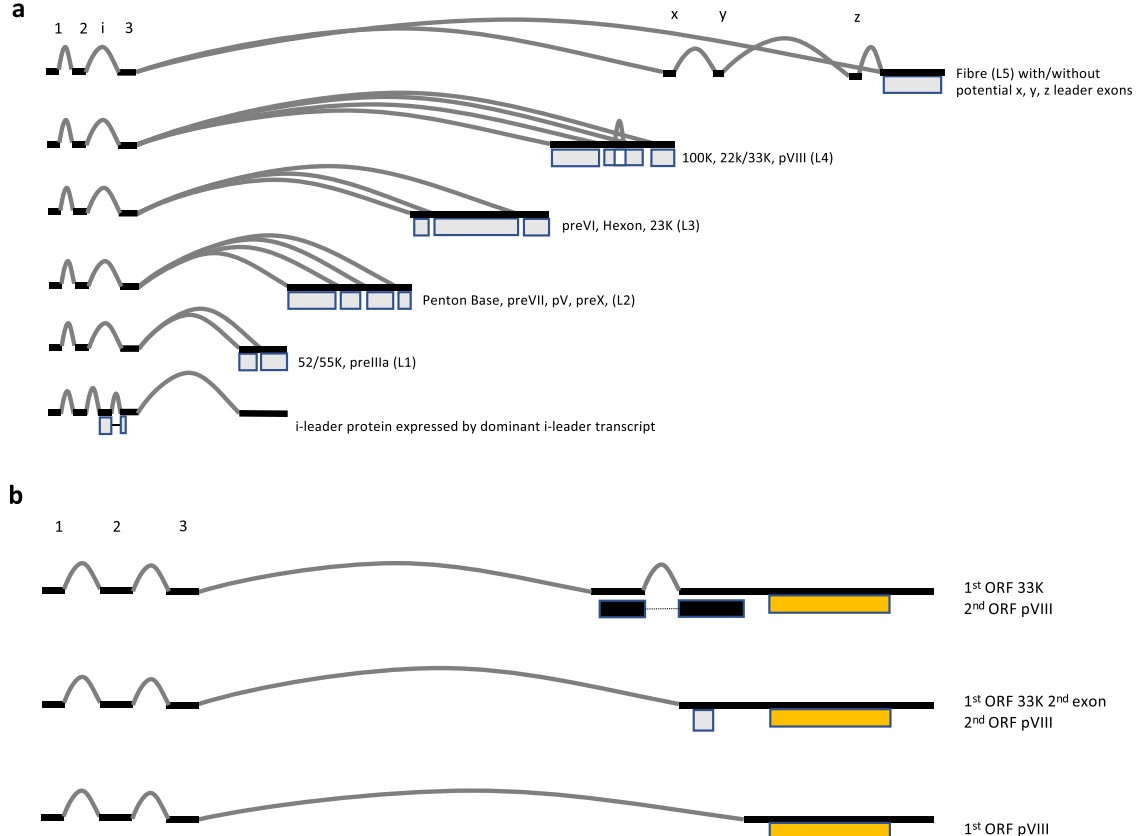

**Fig. 7 Adenovirus major late transcripts. a** The splicing events that give rise to the canonical set of adenovirus MLTU transcripts, grouped by polyadenylation class (L1–L5). Exons are shown as solid black lines with splice events depicted by curved grey lines. Boxes indicate the relative locations of the major ORFs. Splicing, as indicated, places the initiating AUG of each ORF immediately downstream of the TPL (exons 1, 2 and 3). Also shown is the dominant transcript that codes for the i-leader protein, which is in the L1 class. **b** Splicing to generate 33K, 33K 2nd exon/preVIII and preVIII transcripts. The ORF for 33K is shown in black, the theoretical ORF 33K 2nd exon in grey and the preVIII ORF in yellow.

**Major late transcripts containing i-leader, x, y and z exons**. Long-read sequencing allows a quantitative assessment of i-leader exon inclusion into different MLTU mRNA classes for the first time. L1 52/55K transcripts most frequently include an i-leader between TPL2 and TPL3 (ref. [17]), but every major class included the i-leader exon at a measurable rate (Fig. 7a, Supplementary Table 4). Inclusion of i-leader followed by TPL3 forms an ORF that terminates in TPL3 and encodes the i-leader protein regardless of downstream splicing events. However, previously undescribed splicing arrangements following the i-leader exon were readily detected, especially TPL3 exon skipping, which links the i-leader exon and its incomplete ORF with a variety of 3′ exons from L1–L5. We found over 3000 distinct transcripts of this type belonging to 900 different transcript groups creating 60 distinct ORFs, with the i-leader protein N terminus fused to different C termini. Thus, our analysis highlights the i-leader coding transcripts as an under-considered part of the adenovirus gene expression repertoire.

Original analyses of MLTU transcripts using electron microscopy indicated the variable inclusion in L5 fibre-encoding messages of one or more additional leaders x, y and z (refs. [4,37]; Fig. 8a), between the TPL and the fibre-coding exon. Similar to i-leader, inclusion of these exons can now be quantified (Fig. 8b) but this is complicated by the large number of distinct transcript groups found to contain one or more of them, some initiating at the E3 promoter rather than the MLP. Also, the x, y- and z-leader exons have intron–exon boundaries that are integral to expression

of some E3 proteins, meaning they are established components of some E3 mRNAs. To assess leader inclusion in fibre transcripts, we focused on transcripts that derived from MLP and contained one or more of the x-, y- or z-leader exons. Most fibre-encoding transcripts contained none of the x-, y- and z-leaders, with those containing just the y-leader exon being the next most abundant. As has been previously reported, addition of all three x-, y- and z-leaders to a transcript was very rare[4]. Some transcripts included both x- and y-leader exons, and meaning they would express a truncated N-terminal part of the E3 12.5K ORF; this 5′ AUG would need to be ignored to express fibre.

**L4 promoter**. We also examined the role of the intermediate-phase L4 promoter (L4P)[38] in driving expression of the L4 33K and L4 22K genes. These proteins, previously thought to be solely expressed from MLP, are needed to drive late gene expression from the MLP, which presented a paradox until L4P was discovered. At all three time points, both proteins were coded by transcripts starting at either promoter (Supplementary Table 5). However, 33K transcripts were always in much greater abundance than 22K, and at all time points L4P-derived transcripts were predominantly 22K-encoding, whereas 33K-coding transcripts were predominantly from the MLP.

**Polyadenylation length and location**. Nanopore raw sequence data contains information on polyA tail length. There was a consistent trend for shorter polyA tails as the infection progressed

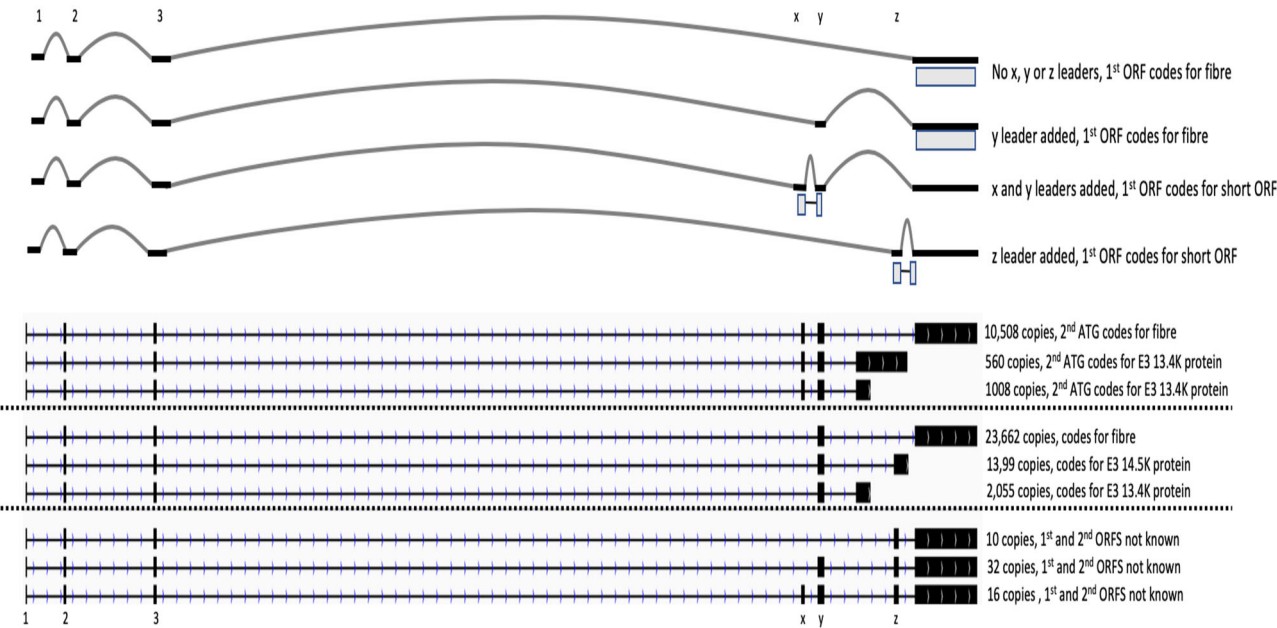

**Fig. 8 Effects of adding x-, y- and z-leaders to the fibre transcript. a** The effects on coding potential of selective combinations of x, y and z exons added to an MLP-derived transcript that would otherwise code for the fibre protein. The three TPL exons of a classical major late transcript are labelled 1, 2 and 3. Grey boxes indicate the relative locations of potential ORFs. **b** Transcript groups detected that correspond to x, y and/or z-leader inclusion into MLP-derived L5 fibre transcripts. The three most abundant transcript groups, separated by dotted lines, that contain either an x-leader exon (b1), a y-leader exon (b2) or a z-leader exon (b3) are shown (image generated by IGV viewer). Coding capacity and transcript numbers are shown at the right of each transcript group representation. Note that in the groups containing the z-leader exon there are multiple ATG (potential initiating) codons in the z-leader which, when spliced as shown, do not lead to a known adenovirus protein being coded.

for most classes of intermediate (pIX and IVa2), late and E2 transcripts (Fig. 9) although with high variance: standard deviation for the average poly-A length was around 50 for most transcript classes (Supplementary Data 4). For MLTU transcripts, the reduction in average polyA length, both 16–24 h.p.i. and 24–48 h.p.i., was highly significant by both Wilcoxon signed-rank test and Mann–Whitney $U$ test, ($p < 0.001$, Supplementary Data 5). Amongst early genes, the E1, E3 and E4 transcripts showed a variable pattern of changes. Inclusion of the i-leader intron was associated with a modest increase in polyA tail length ($p = 9 \times 10^{-4}$; Supplementary Table 4, Supplementary Data 5).

**Splice site detection**. An illumina sequencing dataset is, in effect, an untargeted RT-PCR survey of mRNA splicing. We compared the location of splice sites identified by dRNA-seq with those detected in matched Illumina data: in the majority (88%) of nanopore-derived transcripts, every splice site on each nanopore transcript was also present in the Illumina data (Supplementary Data 6–8). We propose that this is a useful high-throughput validation of the nanopore data, as the Illumina and nanopore data derive from distinct protocols and sequencing technologies.

**Non-classical transcripts**. Multiple transcript types capable of being translated into the same known adenovirus protein were consistently observed. Conversely, many splicing events produced transcripts containing unknown ORFs—between 11% and 16% of detected transcripts did not code for any known adenovirus proteins (Supplementary Data 5). Whether these proteins are made and are functional is beyond the scope of this manuscript, but the large number of alternative transcripts is striking. For example, focusing on fibre-coding transcripts and considering only those that started and terminated at the dominant TSS (nucleotide 6051) and TTS (nucleotide 32,790) sites to exclude

transcript groups that likely arose from truncated transcripts, 123 different groups were found (covering 59,005 independent sequence reads). Even considering only transcript groups with ten or more supporting reads, there were 17 distinct transcript groups with fibre protein as the 5′ proximal ORF (Fig. 10a); similar observations were made for most of the known ORFs (e.g., Figs. 4 and 6). Most notably among fibre-coding transcripts, the TPL was not always intact, with TPL exon 2 being sometimes spliced directly to the y-leader exon or to the fibre ORF; low-frequency omission of TPL exon 3 was also seen in transcripts encoding other late proteins. Previously, the TPL (sometimes with the further inclusion of the i-leader exon, above), has been viewed as an invariant feature of MLP-derived transcripts. Also, none of the most abundant transcript groups contained the x- or z- leader exons, which were previously reported to be found in a proportion of L5 mRNAs (ref. [4]); instead an apparently novel exon was found >50 times in two transcript groups, positioned 3′ to the y-leader exon. Finally, the sizes of TPL exons 2 and 3, as well as the y-leader exon, varied among the groups as a result of usage of splice donor and/or acceptor sites which have not been reported previously.

As stated, we used the Illumina dataset as a large-scale untargeted RT-PCR survey to independently verify the bulk of the splice events. To add further independent verification, a targeted RT-PCR analysis was used to validate four examples of such splice events within previously unreported and rare fibre transcripts (Fig. 10b, Supplementary Fig. 1); in each case, amplified products of the predicted sizes were readily detected.

## Discussion

Understanding viral transcriptomes represent a unique challenge due to the compact nature of their genomes. Adenoviruses use of alternative splicing combined with a modest number of

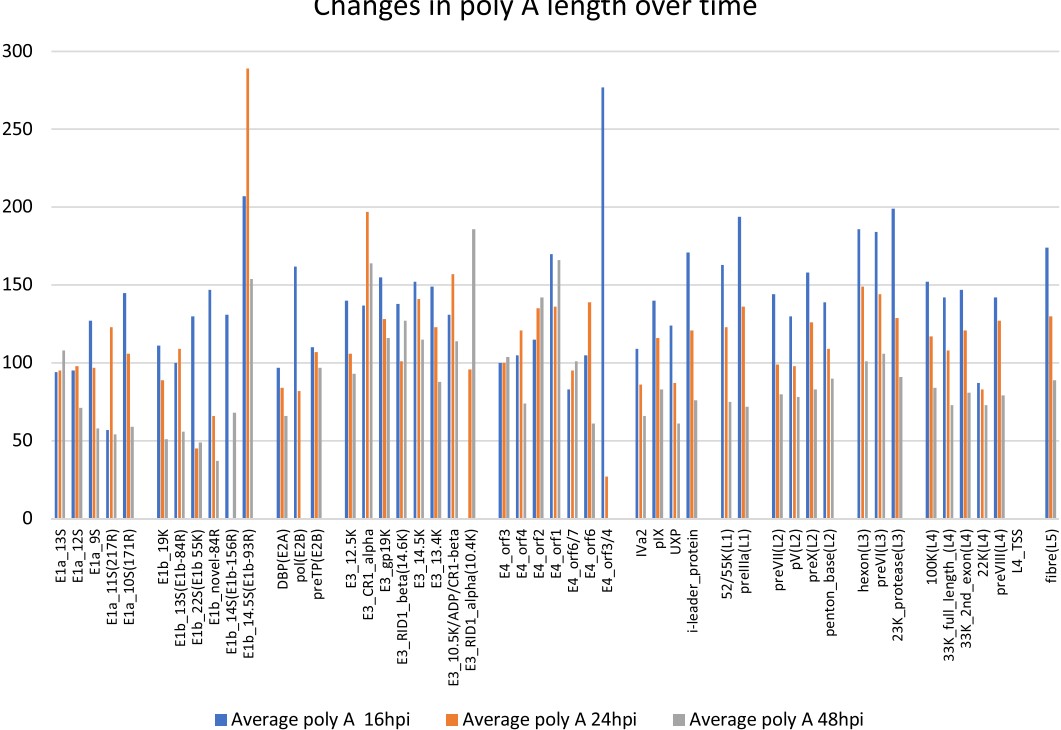

**Fig. 9 Average polyA lengths change over time during adenovirus infection.** This chart shows the changes in mean polyA length over time for the dominant transcript group that codes for each indicated ORF.

promoters and polyadenylation sites makes for a particularly complex transcriptome. Part of this challenge is the large volume of data generated by dRNA-seq. Our approach to managing this data complexity was to group transcripts based on their structure and then to annotate them for their potential to code for known proteins, which in turn allowed definition of suspected novel protein-coding opportunities within the RNA. This supervised, but essentially unbiased, approach was shown to produce a transcript map for human adenovirus that was strikingly similar to the classical map.

This dataset provides the most detailed and quantitative view of the adenovirus type 5 transcriptome yet achieved. For the most part, observed transcript abundances for the major proteins were as expected. However, one (of several) notable exceptions is preX. This is a 79aa protein precursor that is cleaved after virion assembly into three components, the middle 19aa of which comprise the mature protein X that is associated with viral DNA inside the particle alongside viral histone-like proteins pV and pVII (refs. [39–41]). However, the roles of the other portions of preX that are liberated by this cleavage and any additional roles of the precursor relative to the mature protein are not understood. Although the excess of preX transcripts over those of pV and preVII may be needed to compensate for differences in translation rates and stability of these proteins, we believe it also suggests additional role(s) for preX. Indeed, there are many cases where our observations raise interesting questions in relation to the large body of previously published work on adenoviruses. While it is beyond the scope of this manuscript to compare in detail this dataset with that work, we believe that the raw and analysed data will prove useful to other teams working on adenovirus transcription and splicing control.

A key aspect of our analysis is the diversity and relative abundance of distinct mRNAs that effectively code for the same protein. This may permit protein production to be fine-tuned through alteration in the balance between different mRNA groups

expressing that ORF. We propose that we are observing this process directly with pVIII being coded both by a "classical" transcript and potentially by the second ORF on the more abundant 33K 2nd exon transcript but with lower intrinsic efficiency. Although there is no reliable method of predicting how efficiently any given AUG will be used[42], adenovirus uses secondary AUGs as initiation codons for most E1b proteins and for some E3 proteins, and translation termination/re-initiation on an mRNA has been reported in respiratory syncytial virus at least[27]. Our data reinforce the idea that non-standard translation initiation may extend further across adenovirus's transcriptomic repertoire. Further work is underway to try to detect expression from these apparently novel ORFs predicted by our data.

dRNA-seq technology has great potential application but interpreting such sequencing data still requires careful analysis, including when determining the 5′- and 3′-ends of transcripts. Adenoviral TSSs have been mapped previously with high precision which, coupled with the very high abundance of viral transcripts, allows rigorous assessment of the accuracy of 5′-end position determined from the nanopore data. The 5′-end position inferred by dRNA-seq was 11–16 nt downstream of the previously mapped 5′-end for all the adenovirus promoters—a number that agrees with human dRNA-seq datasets[14]. The precise peak count position at each promoter was reproducible between different time point samples, indicating that differences in the sequence near the 5′-end affect how close to the end of an RNA molecule dRNA-seq can read. At the 3′-end there are distinct issues too; even though we could both detect and measure the length of poly-A tail, there was a notable lack of precision over the TTS. That the mapping software usually clipped the 3′-end of a sequence when aligning it to the genome suggested different possibilities. Potentially, mapping accurately within what is typically an A/U rich region poses problems for mapping algorithms. Alternatively, as the raw nanopore signal transitions from poly-A tail to the mixed nucleotide sequence of the RNA,

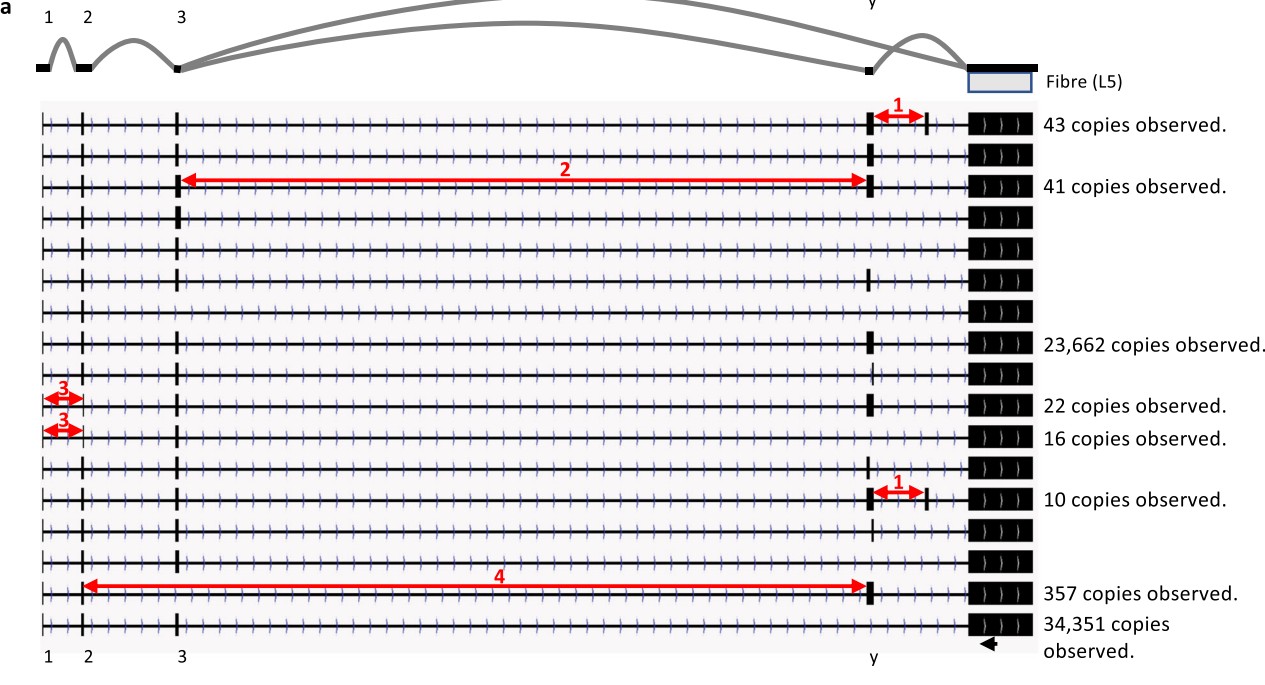

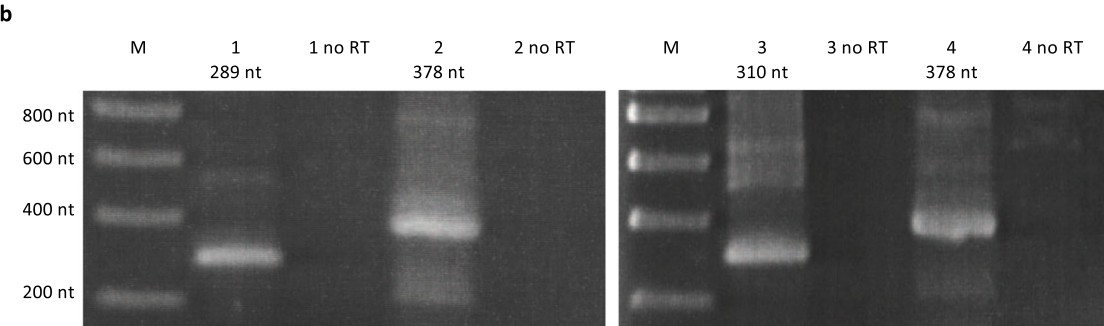

**Fig. 10 Distinct transcripts coding for fibre protein. a** shows the 17 transcript groups having at least ten observed transcripts, in which the fibre ORF was the first ORF are represented as lines with exons as black boxes. The numbers of transcripts belonging to the three most abundant transcript groups and the transcripts selected for targeted RT-PCR validation is shown to the right; the remaining transcripts were observed between 10 and 72 times. The exon structure and splicing pattern of the two most abundant transcript groups is shown schematically at the top. The locations of the tripartite leader exons and the y-leader exon are also shown. Note that in many cases the transcripts have truncated y-leader exons and in some cases, there is a novel exon downstream of the y-leader exon (i.e., it is not the x-leader exon). In order to validate by targeted PCR, we designed four primers that span two uniquely connected exons. Which exon pairs are spanned by each primer are numbered and indicated in the diagram by connecting double-headed red arrows. We also designed a universal reverse primer for the fibre ORF some 180 nt upstream of the fibre start codon and indicated by a black arrow at the bottom of the figure in **a**. **b** shows the results of RT-PCR (with and without reverse transcription) using the universal reverse primer and the numbered forward primers indicated in **a**—the expected size of the PCR product is indicated for each primer pair. Also indicated are the marker lanes (M) and the size of the markers are shown on the left hand side.

there may be additional difficulties with accurate base calling. The shortening of the viral poly-A tail for viral late transcripts is intriguing; a similar observation has been reported for coronaviruses and this was associated with reduced translation efficiency[43]. Further work will be needed to see if this is the case for adenovirus. We propose that adenovirus is a useful testbed for dRNA-seq experiments aimed at improving the capture of 5'- and 3'-end sequence information.

As part of the detailed dRNA-seq analysis of the adenovirus transcriptome, we developed new software tools. This software pipeline should permit others to examine not just different viruses via this approach but also different strains of the same virus. It should allow rapid characterisation of the transcriptional and coding landscape of a broad range of viral pathogens.

Moreover, the principles of our analysis pipeline could be applied beyond viral systems.

The data also offer further insight into how adenoviruses, and perhaps DNA viruses generally, can evolve. DNA viruses are traditionally perceived as relatively static in evolutionary terms compared to RNA viruses. Our analysis shows that human adenovirus type 5 produces, at low frequency compared to the well-established repertoire of RNAs, transcripts with an astonishing array of exon combinations, just over 11,000 across all three time points. To provide confidence that these thousands of splice events were genuine, we independently validated them by comparison of splice sites inferred by the nanopore data and by Illumina data. We also validated four of these previously unreported splice events by targeted RT-PCR analysis.

We speculate that flexibility in RNA processing enables exploration of the fitness benefit of novel exon combinations. Production of beneficial exon combinations could yield a selectable advantage. Splice site selection is influenced by many factors, including the sequences surrounding the splice acceptor–donor sites[44]; random changes that favoured increased use by the host splicing apparatus of exon combinations already in low-level use would be a mechanism for favouring an advantageous splice site combination.

Considering splicing variation and the use of secondary start codons together, our genome-wide overview suggests that, far from having a relatively static coding capacity, adenovirus is routinely producing different combinations of splice acceptor–donor pairs and using secondary start codons at low or modest levels. Over time, one such combination could enable expression of a protein with a selectable advantage. With this perspective, the use of a small number of promoters but with a wide array of splice acceptor and donor sites, coupled with an active use of secondary start codons, makes adenovirus highly plastic at the level of transcription and translation control.

The frequent use of variant splice acceptor–donor site pairs by adenovirus has implications for adenovirus-based gene delivery vectors, which are widely used both in research and clinical settings. Our data suggest that the full transcriptional repertoire of such vectors should be assessed to ensure that they do not suffer from inadvertent post-transcriptional processing leading to expression of unintended additional or alternative proteins. This is especially true for viral vectors that use the E3 region for transgene expression because this region appears particularly susceptible to variant splicing patterns. Such analysis could also drive improvements in transgene expression control in adenovirus-based vectors.

## Methods

**Virus and cells**. Human MRC-5 cells, a genetically normal human lung fibroblast-like line, were obtained from European Collection of Authenticated Cell Cultures (#05072101, ECACC). The cells were cultured in DMEM supplemented with 10% foetal bovine serum, 100 U/ml penicillin and 100 μg/ml streptomycin. After reaching confluence, the cells were infected with adenovirus type 5 at a multiplicity of 100 fluorescent focus-forming units per cell to ensure infections were as synchronous as possible. The infected cells were harvested at 16, 24 and 48 h.p.i.

**RNA extraction and sequencing**. Total RNA was extracted from the infected cell using TRIzol™ reagent (#15596026, Ambion) at 1 ml of reagent per $10^7$ cells and the RNA extracted as per manufacturer's recommendations except that the final wash of the RNA pellet in 70% ethanol was repeated a further two times (total of three washes). At this stage, the RNA could be stored in 70% ethanol for at least two weeks at $-80\,^\circ$C without degradation. Once the ethanol-washed RNA was precipitated and air dried it was resuspended in RNAse-free water, and taken through the polyA enrichment and RNA-seq protocol as rapidly as possible without pausing or storage. The additional ethanol washes and rapid processing had a major impact on the quantity and quality of the sequence data returned. Initially the RNA was assessed for purity and quantity using nanodrop, and RNA was only used if the A260/A280 ratio was 2.0 ± 0.05 and the A260/A230 ratio was >2 and <2.2. Total RNA was enriched for polyA tails using Dynabeads™ mRNA purification kit (#61006, Invitrogen) as per manufacturer's instructions, typically inputting 40–50 μg of total RNA. After enrichment, the quantity of mRNA was assessed again using nanodrop and 400–500 ng of RNA was used for sequencing. We used the SQK-RNA002 kits and MIN106D R9 version flow cells (Oxford Nanopore Technologies), following the manufacturer's protocols exactly. We typically obtained 1.5–2.2 million QC-passed reads per flow cell over 48 h, notably more than has been published to date[14].

**Targeted novel splice site validation**. We designed primers that crossed a range of novel exon–exon boundaries found in the fibre transcripts as well as a universal reverse primer within the fibre open reading frame (supplementary PCR methods). These were used in a targeted RT-PCR experiment with the SuperScript III RT-PCR kit (Invitrogen) and the PCR products analysed on agarose gels.

**Data analysis, error correction and mapping to viral genome**. Reads were initially mapped to the human genome using minimap2 (ref. [45]) to filter out host transcripts and reads that did not map were then converted back to fastq files using an in-house script. These unmapped reads were then presumed to be mostly viral and the sequence errors in the raw data, which are inherent with nanopore sequencing, were corrected using LoRDEC (ref. [46]) software and accurate Illumina-derived short-read sequences that had been previously generated using the same batch of adenovirus[8] as used here (command line: lordec-correct -T 4 -k 19 -s 3 -a 10000–2 adenovirus_illumina_reads.fastq -i adenovirus_nanopore_reads.fastq -o adenovirus_nanopore_reads_corrected_by_illumina_K19.fasta). We used iterative rounds of correction on the nanopore data with increasing K-mer values as recommended, in this case we repeated each round of correction before increasing the K-mer values (e.g., K19, K19, K31, K31, K41, K41 and finally K51, K51). This correction is needed to improve the accuracy of mapping of the corrected transcripts to the viral genome, which was done using minimap2 with pre-set values for long-read spliced transcripts (command line: minimap2 -ax splice -uf -k14 –sam-hit-only adenovirus_genome.fasta corrected_nanopore_reads.fasta > nanopore_reads_mapped_to_virus.sam). This generated >1.1 million mapped adenovirus reads in total, split between the three time points.

**Data analysis, characterisation of viral transcripts**. To cope with the very wide range of transcripts and to enable grouping of transcripts into classes, two in-house scripts were developed to count and classify the mRNA. The first (classify_transcripts_and_polya.pl) establishes the locations of TSS and TTS alongside possible splice acceptor/donor sites. This script produces tables indicating where on the genome and how often in the data each TSS, TTS or splice site occurs. Subsequently, these events are grouped for simplicity of analysis. For example, the most frequently used polyadenylation site is established and all other polyadenylation sites within a user defined window (in our case 15 nucleotides either side) are presumed to be grouped with that location. Once this is done the next most abundant site outside this 15-nucleotide window is selected and so on until all the polyadenylation sites are accounted for. In this analysis, splice acceptor and donor sites are not grouped. Once this is complete, the software then assigns each transcript to a "transcript group" depending on its pattern of TSS, splice sites and TTS locations, and counts how many transcripts belong to each transcript group.

Allied to this analysis, nanopolish[14] was used to determine the polyA length of each sequenced transcript and subsequently the average polyA length was calculated for each transcript group. For this analysis, only transcripts that were QC-flagged as PASS by nanopolish polyA (~60% of transcripts) and had an estimated polyA tail length of 20 or more were considered.

An in-house script (name_transcripts_and_track.pl) determines which known features are present in each transcript group. It first uses the transcription start, termination and splice site locations for each transcript group to create a pseudo transcript based on the genome sequence, in effect a consensus for the group. The user can specify how many nucleotides upstream of the transcript start site for the group should be included in each pseudo transcript, in order to compensate for any observed failure to fully sequence the 5′ ends of the RNA (see Results section). The script then examines each pseudo transcript to determine what features it has, using a user-specified list of canonical features or ORFs on the viral genome (Supplementary Data 9). For example, it will define the 5′ most ORF and if that ORF is not canonical, search for a canonical ORF coded by the next available start codon. Finally, the software produces tables that indicate how many transcripts in total, from all the transcript groups, code for each ORF listed in the features table. The tables also include additional data, such as average polyA length for each transcript group, and the sequences of the splice donor and splice acceptor sites for all exons detected. It also produces GFF files that allow the user to visualise the dominant transcript coding for each ORF, as well as GFF files that describe the whole range of transcripts coding for any given ORF (Supplementary Data 10–13). In addition, an analysis counting the final number of transcripts belonging to each translated feature is produced (Supplementary Data 14). This produces an ORF-centric view of the viral transcriptome—classifying transcripts according to which proteins they could produce, if translated.

To achieve high-throughput verification of the large number of splice sites determined by direct mRNA-seq on the nanopore, we used the illumina data as an independent, large scale and unbiased RT-PCR survey of splicing events in the mRNA sample. The illumina data were first mapped to the adenovirus genome using HISAT2 (ref. [47]) before using an in-house script (compare_nanopore_splices_to_illumina.pl) to determine which splice events identified by the nanopore data were also present in the illumina data.

**Statistics and reproducibility**. The primary data were circa 1.2 million viral transcript reads from nanopore dRNA-seq analyses of adenovirus-infected cell RNA at different times post infection, each condition as a single biological replicate. For analysis of differences in mean polyA length over time among MLTU transcript classes, the number of such classes was defined by the virus transcription map = 15. Differences in paired mean polyA length between time points analysed by Wilcoxon ranked-sum test with each mRNA class providing a paired data point, or as unpaired values by Mann–Whitney $U$ test. No other statements have been made about the significance of differences in abundance of specific RNA transcripts between time points that would require statistical testing.

**Reporting summary**. Further information on research design is available in the Nature Research Reporting Summary linked to this article.

## Data availability

The sequence data has been deposited at ENA with the accession numbers ERS3781043, ERS3781044, ERS3781045, ERS3886435 and ERS3781046.

Data is also available from the authors on request or from Zenodo[48], https://doi.org/10.5281/zenodo.3610257.

## Code availability

The software is available from the authors on request or from Zenodo[49], https://doi.org/10.5281/zenodo.3610249.

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

## Acknowledgements
We'd like to thank the BBSRC for funding (D.A.M. and I.D.-B., grant BB/M02542X/1).

## Author contributions
I.D.-B. carried out the laboratory work in this article. I.D.-B., A.S.T., J.A.H., K.N.L. and D.A.M. conceived and designed the program of research, contributed intellectually to drafting the research text and critiqued the intellectual content. K.N.L. provided statistical analysis and tables. D.A.M. supervised the work, wrote the software and used it to analyse the data.

## Competing interests

The authors declare no competing interests.
