## [Peer Review File · Communications Biology]

Reviewers' comments:

Reviewer #1 (Remarks to the Author):

The manuscript by Donovan-Banfield and colleagues describes the use of nanopore MinION RNA long read sequencing to re-examine the transcripts produced by adenovirus during an infection time course. At first sight, this endeavour may seem of limited interest or impact, but in my opinion this is far from the case. The work that the authors describe represents a significant step forward in the application of an important new sequencing technology to quantitatively measure the transcripts generated over time in a well-studied biological system with complex transcription and splicing patterns. Using the adenovirus infection system enabled comparisons to be made with data generated using a wide number of complementary techniques over many decades to map transcript starts and ends and splicing patterns. The detailed analysis and mapping of the mRNA sequenced has advanced progress in the use of this sequencing method and provided analysis pipelines and scripts that will be very useful to researchers across all disciplines interested in direct transcript sequencing and mapping. In terms of biological discoveries, this study has identified new transcript variants, some produced at low level, but others representing a significant proportion of transcripts. It has uncovered previously unappreciated complexity in the generation of multiple differentially-spliced transcripts containing the same ORF indicating that adenovirus (and potentially other DNA viruses) has many ways of maintaining its viral gene expression profile to give it a higher level of adaptability and flexibility than previously observed. Issues with quantitative differential transcript mapping and bias from cDNA generation and PCR amplification with widely-used deep sequencing methods have no doubt prevented the previous observation of this level of complexity. Direct transcript sequencing has also highlighted the likely use of downstream AUG codons and provided insight into changing poly A tail lengths during the course of infection.

The data and new concepts uncovered here will no doubt be of interest to those in the adenovirus and DNA virus field and would be expected to be re-analysed by many groups providing a valuable community resource. Given that the observations also shed light on the fundamental processes of splicing and translation they will also be of broader interest.

Data analysis appears robust and transcript counts are displayed throughout. Data presentation is good overall, considering how complex splicing and transcription patterns are.

Specific comments:

1. Soft-clipping needs more explanation for the non-expert. What is the significance of the observations related to this in the paragraph on p9, line 3.
2. Is there any proof that the downstream AUGs are used for the newly sequenced transcripts where they would be the only way the ORF could be expressed e.g. for DBP. Are they in an appropriate context to be recognised? Is the 5' AUG sub-optimal? Is there any evidence that the short upstream ORF is expressed?
3. Is there any evidence in other viral systems for shortening of poly A tails. What could be the potential impact of this for adenovirus? Do viral mRNA translation rates change during infection?
4. How significant is the presence of 1 or 2 copies of an alternative transcript in the context of many thousands of other transcripts? Could some of these represent background noise as a result of splicing errors?

Reviewer #2 (Remarks to the Author):

The manuscript by Donovan-Banfield and coworkers very nicely describes the application of a relatively new technology, direct RNA long read sequencing on a Oxford Nanopore MinION, to determine the transcriptome of Ad5. The technology has specific advantages for a study like this, but

also weaknesses, and the authors openly acknowledge the potential artifacts in their data. In particular, the arbitrary addition of 10 nucleotides to the end of each transcript was concerning but appears to have allowed the authors to correlate their results with known Ad ORFs. The results of their study suggests that Ad gene expression (transcription, at least) is "messy", something this reviewer thinks has not been shown to this degree previously, hence the novelty of this work.

RNA splicing represents an important source of diversity in response to host cell pressures. However, although the authors may consider it outside the scope of this manuscript, they should, at least in the discussion, better address the impact of this diversity on translation. Not all transcripts are translated, and of those that are, their leader sequences can have a major impact on translation efficiency. If translation is selective, what is the impact on the cell of untranslated transcripts? More importantly, it is also not clear how message diversity might impact the physical virion. If even some of the reported fiber transcripts are translated, would their protein products properly "fit" (bind) to the penton base protein? And if there is variation in the fiber protein, does that mean that individual virions will show variation between their 12 fiber trimers at the aa level? Would variable fiber proteins even properly assemble? In this context, it would have been very impactful if the authors had shown fiber protein variants on the physical virion. And if multiple transcript variants were indeed translated and bound properly to the penton base protein, what is the potential impact on host cell binding and fitness by projected variations in the fiber knob?

Therefore, this reviewer would have liked to have seen some attempt at validation of at least a few of their transcriptome predictions. For example, could the authors have performed 5'RACE and Northern blots to confirm the fiber mRNA variants? Is there any evidence for fiber protein variation for Ad5?

Finally, can the authors address how splicing diversity would lead to genome evolution, i.e. by what feedback mechanism?

Minor points:

Abstract, last sentence: "constitutive low" - did the authors mean "constitute high"?

Please explain the choice of an MOI of 100.

Page 10, line 37, "know" should be "known"

Page 12, line 21. Please clarify that these are p values, and confirm the data showed a normal distribution.

Reviewer #3 (Remarks to the Author):

In this study, the authors used direct RNA long read length sequencing to examine the populations of human adenovirus type 5 mRNA during an infection, which revealed an extensive complexity of alternative splicing and secondary initiating codon usage. This work shows some novelty and academic value. This manuscript needs minor revision before it can be acceptable for Communications Biology.

A list of major concerns is as follows:

1. Is there any technological improvement in this work compared to the previously reported direct RNA sequencing?
2. The authors should provide some related experimental results to further demonstrate the reliability

of data analysis on gene alternative splicing.

3. There are some inaccurate and confusing English usages in the manuscript. The authors should revise their paper carefully.

Re manuscript COMMSBIO-19-1463-T

Responses to the reviewers comments are highlighted and in red.

Reviewers' comments:

Reviewer #1 (Remarks to the Author):

The manuscript by Donovan-Banfield and colleagues describes the use of nanopore MinION RNA long read sequencing to re-examine the transcripts produced by adenovirus during an infection time course. At first sight, this endeavour may seem of limited interest or impact, but in my opinion this is far from the case. The work that the authors describe represents a significant step forward in the application of an important new sequencing technology to quantitatively measure the transcripts generated over time in a well-studied biological system with complex transcription and splicing patterns. Using the adenovirus infection system enabled comparisons to be made with data generated using a wide number of complementary techniques over many decades to map transcript starts and ends and splicing patterns. The detailed analysis and mapping of the mRNA sequenced has advanced progress in the use of this sequencing method and provided analysis pipelines and scripts that will be very useful to

researchers across all disciplines interested in direct transcript sequencing and mapping.

In terms of biological discoveries, this study has identified new transcript variants, some produced at low level, but others representing a significant proportion of transcripts. It has uncovered previously unappreciated complexity in the generation of multiple differentially-spliced transcripts containing the same ORF indicating that adenovirus (and potentially other DNA viruses) has many ways of maintaining its viral gene expression profile to give it a higher level of adaptability and flexibility than previously observed. Issues with quantitative differential transcript mapping and bias from cDNA generation and PCR amplification with widely-used deep sequencing methods have no doubt prevented the previous observation of this level of complexity. Direct transcript sequencing has also highlighted the likely use of downstream AUG codons and provided insight into changing poly A tail lengths during the course of infection.

The data and new concepts uncovered here will no doubt be of interest to those in the adenovirus and DNA virus field and would be expected to be re-analysed by many groups providing a valuable community resource. Given that the observations also shed light on the fundamental processes of splicing and translation they will also be of broader interest.

Data analysis appears robust and transcript counts are displayed throughout. Data presentation is good overall, considering how complex splicing and transcription patterns are.

Specific comments:

1. Soft-clipping needs more explanation for the non-expert. What is the significance of the observations related to this in the paragraph on p9, line 3.

A: We have added this on page 9 (lines 4-8) and discussed the implications p15 (lines 12-17).

2. Is there any proof that the downstream AUGs are used for the newly sequenced transcripts where they would be the only way the ORF could be expressed e.g. for DBP. Are they in an appropriate context to be recognised? Is the 5' AUG sub-optimal? Is there any evidence that the short upstream ORF is expressed?

A: We make no claims that novel proteins are being made and the criteria for an AUG to be used for initiation are too vague to allow such prediction with any real certainty. We have emphasised this in the text on page 14 (lines 35-45).

3. Is there any evidence in other viral systems for shortening of poly A tails. What could be the potential impact of this for adenovirus? Do viral mRNA translation rates change during infection?

A: We have found research that directly investigates the variation in poly A tail length in coronavirus infected cells over time. This suggests that shortening poly A length leads to less efficient translation. We have added a short commentary to that effect on page 15 (lines 17 - 20).

4. How significant is the presence of 1 or 2 copies of an alternative transcript in the context of many thousands of other transcripts? Could some of these represent background noise as a result of splicing errors?

A: We do believe that the rare alternative transcripts are evidence of a "sloppy" splicing system which could permit evolution of the virus over time. There is in fact no viable distinction between a rare 'correct' splicing event and an 'error' by the splicing apparatus since there is no external definition of what is 'correct' splicing for any transcript; the novel splice events we detect occur at sites that match the basic features of such sites, suggesting the events do utilise the splicing apparatus rather than being random breakage – re-joining products. We have checked the text again to make sure this point is clear and have added an additional short statement and reference to splice site selection on page 15 (lines 37-40).

Reviewer #2 (Remarks to the Author):

The manuscript by Donovan-Banfield and coworkers very nicely describes the application of a relatively new technology, direct RNA long read sequencing on a Oxford Nanopore MinION, to determine the transcriptome of Ad5. The technology has specific advantages for a study like this, but also weaknesses, and the authors openly acknowledge the potential artifacts in their data. In particular, the arbitrary addition of 10 nucleotides to the end of each transcript was concerning but appears to have allowed the authors to correlate their results with known Ad ORFs. The results of their study suggests that Ad gene expression (transcription, at least) is "messy", something this reviewer thinks has not been shown to this degree previously, hence the novelty of this work.

RNA splicing represents an important source of diversity in response to host cell pressures. However, although the authors may consider it outside the scope of this manuscript, they should, at least in the discussion, better address the impact of this diversity on translation. Not all transcripts are translated, and of those that are, their leader sequences can have a major impact on translation efficiency. If translation is selective, what is the impact on the cell of untranslated transcripts? More importantly, it is also not clear how message diversity might impact the physical virion. If even some of the reported fiber transcripts are translated, would their protein products properly "fit" (bind) to the penton base protein? And if there is variation in the fiber protein, does that mean that individual virions will show variation between their 12 fiber trimers at the aa level? Would variable fiber proteins even properly assemble? In this context, it would have been very impactful if the authors had shown fiber protein variants on the physical virion. And if multiple transcript variants were indeed translated and bound properly to the penton base protein, what is the potential impact on host cell binding and fitness by projected variations in the fiber knob?

A: Again, we make no claims at all that there are any novel proteins being made. While there are many many transcript variants, all code for the same fiber protein. The key point here is that the transcripts exist and if any provided a selective advantage then they would, over time, have begun to dominate. As before we have revised the wording throughout the text to make sure this is clear.

Therefore, this reviewer would have liked to have seen some attempt at validation of at least a few of their transcriptome predictions. For example, could the authors have performed 5'RACE and Northern blots to confirm the fiber mRNA variants? Is there any evidence for fiber protein variation for Ad5?

A: We do feel that the illumina data itself is an untargeted RT-PCR validation of our findings using direct RNA sequencing and we have strengthened this in the text (e.g. page 15 lines 37-40 and page

13 lines 16-20). We accept the suggestion that further proof is useful and we have examined a few novel transcripts by targeted RT-PCR as additional assurance that the novel transcripts exist. We have provided additional targeted RT-PCR data in figure 10.

Finally, can the authors address how splicing diversity would lead to genome evolution, i.e. by what feedback mechanism?

A: We have added a short section in the discussion pointing out that SNPs in and around the splice acceptor-splice donor sites would influence how and when a splice site pair would be used (otherwise, clearly, all GU-AG pairing would be used equally) and that this would be the feedback mechanism on page 15 (lines 44-47).

Minor points:

Abstract, last sentence: "constitutive low" - did the authors mean "constitute high"?

A: No, we do mean constitutive low. We propose that the ongoing low level use of alternative splicing events allows the virus to explore its full genetic potential over evolutionary timescales by testing the usefulness of alternative splicing arrangements.

Please explain the choice of an MOI of 100.

A: To ensure in this slow growing cell line that we obtained an almost synchronous infection, we have amended the text on page 5 (lines 7 and 8) to clarify this.

Page 10, line 37, "know" should be "known"

A: Changed as requested.

Page 12, line 21. Please clarify that these are p values, and confirm the data showed a normal distribution.

A: These were p values derived from a t-test. Whilst we believe that the underlying data (polyA length on individual transcripts) are normally distributed, making this test applicable, the small size of the dataset of means (15 paired values) makes it difficult to demonstrate a normal distribution formally. We have therefore replaced the t-test data with results from equivalent non-parametric tests, treating the data as paired for each transcript type (Willcoxon) or unpaired, ie by time point (Mann-Whitney). The difference in average polyA length over time that we highlighted remains highly significant by either test; this has been clarified in the text on page 12 (lines 23-28).

Reviewer #3 (Remarks to the Author):

In this study, the authors used direct RNA long read length sequencing to examine the populations of human adenovirus type 5 mRNA during an infection, which revealed an extensive complexity of alternative splicing and secondary initiating codon usage. This work shows some novelty and academic value. This manuscript needs minor revision before it can be acceptable for Communications Biology.

A list of major concerns is as follows:

1. Is there any technological improvement in this work compared to the previously reported direct RNA sequencing?

A: The paper does not directly concern itself with trying to improve the nanopore technology although we do make the point that our protocol for mRNA extraction and sequencing offers much higher levels of data return per flow cell than any other manuscript we know of (including one published in November in Nature Methods). We have added this point to the text p5, lines 28-29.

The other advances are in the data analysis and transcript characterisation.

2. The authors should provide some related experimental results to further demonstrate the reliability of data analysis on gene alternative splicing.

A: As for reviewer 2 we have provided some direct targeted RT-PCR validation as well as providing more clarity over the utility of illumina data as an independent method of large scale splice site verification (p7 lines 1-3).

3. There are some inaccurate and confusing English usages in the manuscript. The authors should revise their paper carefully.

A: We have revised carefully throughout.

REVIEWERS' COMMENTS:

Reviewer #1 (Remarks to the Author):

The authors have answered my comments and questions satisfactorily and have amended the manuscript as required.

Reviewer #2 (Remarks to the Author):

This reviewer appreciates the clear and constructive responses to the submitted comments.

Regarding: " The shortening of the length of viral poly A tail for viral late transcripts 546 is intriguing, a similar shortening of poly A tail length over time has been reported for 547 coronaviruses and has been associated with a reduced translation efficiency⁴⁶." line 546. Please revise to either two sentences or use a semicolon to separate.

Reviewer #3 (Remarks to the Author):

Thanks. I have no question.